# Is There a Role for Immunotherapy in Prostate Cancer?

**DOI:** 10.3390/cells9092051

**Published:** 2020-09-08

**Authors:** Alessandro Rizzo, Veronica Mollica, Alessia Cimadamore, Matteo Santoni, Marina Scarpelli, Francesca Giunchi, Liang Cheng, Antonio Lopez-Beltran, Michelangelo Fiorentino, Rodolfo Montironi, Francesco Massari

**Affiliations:** 1Oncologia Medica, Azienda Ospedaliero-Universitaria di Bologna, via Albertoni, 40138 Bologna, Italy; rizzo.alessandro179@gmail.com (A.R.); veronica.mollica7@gmail.com (V.M.); 2Section of Pathological Anatomy, School of Medicine, Polytechnic University of the Marche Region, United Hospitals, 60126 Ancona, Italy; alessiacimadamore@gmail.com (A.C.); m.scarpelli@univpm.it (M.S.); r.montironi@staff.univpm.it (R.M.); 3Oncology Unit, Macerata Hospital, 62012 Macerata, Italy; mattymo@alice.it; 4Department of Pathology, Ospedale Maggiore and University of Bologna, 40138 Bologna, Italy; frachikka@virgilio.it (F.G.); michelangelo.fiorentino@unibo.it (M.F.); 5Laboratory Medicine and Department of Pathology, Indiana University School of Medicine, Indianapolis, IN 46202, USA; 6Department of Surgery, Cordoba University Medical School, 14071 Cordoba, Spain; em1lobea@gmail.com

**Keywords:** prostate cancer, immunotherapy, pd-1, CTLA-4, predictive biomarkers, vaccines, immune checkpoint inhibitors, combination therapy

## Abstract

In the last decade, immunotherapy has revolutionized the treatment landscape of several hematological and solid malignancies, reporting unprecedented response rates. Unfortunately, this is not the case for metastatic castration-resistant prostate cancer (mCRPC), as several phase I and II trials assessing programmed death receptor 1 (PD-1) and cytotoxic T-lymphocyte antigen-4 (CTLA-4) inhibitors have shown limited benefits. Moreover, despite sipuleucel-T representing the only cancer vaccine approved by the Food and Drug Administration (FDA) for mCRPC following the results of the IMPACT trial, the use of this agent is relatively limited in everyday clinical practice. The identification of specific histological and molecular biomarkers that could predict response to immunotherapy represents one of the current challenges, with an aim to detect subgroups of mCRPC patients who may benefit from immune checkpoint monoclonal antibodies as monotherapy or in combination with other anticancer agents. Several unanswered questions remain, including the following: is there—or will there ever be—a role for immunotherapy in prostate cancer? In this review, we aim at underlining the failures and promises of immunotherapy in prostate cancer, summarizing the current state of art regarding cancer vaccines and immune checkpoint monoclonal antibodies, and discussing future research directions in this immunologically “cold” malignancy.

## 1. Introduction

Prostate cancer (PC) is the most commonly diagnosed cancer in men, representing one of the leading causes of cancer-related death worldwide [1]. Although patients with localized disease are typically treated with definitive therapy (prostatectomy or radiotherapy, or both), up to 40% of subjects receiving radical prostatectomy (RP) and up to 50% of patients receiving radiotherapy will experience recurrence of disease [2]. Consequently, many patients affected by metastatic disease will develop metastatic castration resistant PC (mCRPC) [3]. Due to the improved knowledge in terms of molecular mechanisms underlying progressive disease and metastatic onset, in the past two decades we have witnessed a considerable increase in the number of therapeutic options for mCRPC, with several agents entered into everyday clinical practice, including docetaxel, cabazitaxel, abiraterone, enzalutamide, and radium-223 [4,5,6,7,8].

Unfortunately, immune checkpoint monoclonal antibodies are not included among these drugs. In fact, although modern immunotherapy has revolutionized the management of a number of malignancies, immune checkpoint monoclonal antibodies are still looking for their niche in several tumors where these agents do not seem to provide ideal results in unselected patients—such as mCRPC [9]. Despite FDA approving sipuleucel-T in 2010 [10], thus becoming the first immunotherapy for mCRPC, recent trials assessing the role of immune checkpoint monoclonal antibodies have shown disappointing results; so far, the only approved immune checkpoint monoclonal antibody is the anti-PD-1 agent pembrolizumab, which can be used in the subgroup of mCRPC patients with high microsatellite instability (MSI-H) [11]. If PD-L1 status, MSI-H, and other biomarkers may identify a subset of patients who are most likely to respond, improving the precision in order to select the responders is a major goal [12].

In this review, we summarize the current state of art regarding immunotherapy in PC, including biomarkers of response, cancer vaccines, chief trials on immune checkpoint monoclonal antibodies and novel immune-based combinations in this immunologically “cold” malignancy with many unanswered questions. We performed a research on Scopus, Cochrane library, and Pubmed/Medline using the keywords “prostate cancer” OR “metastatic castration resistant prostate cancer” AND “cancer vaccines” OR “immunotherapy” OR “immune checkpoint inhibitors”. We selected the most relevant and pertinent studies assessing immunotherapy in PC. Lastly, we performed a research on the clinicaltrials.gov database for recruiting and active, not recruiting, trials.

## 2. Predictive Biomarkers: PD-L1, MSI, MMR, TMB, DDR, TILs

The last decade has seen outstanding improvements in medical oncology, with the development and emergence of several novel agents and combinations [13,14,15]. Among these therapeutic approaches, a key role has been played by immune checkpoint monoclonal antibodies which have reported noteworthy results in a wide number of malignancies [16,17,18]. For example, medical treatment of metastatic melanoma, urothelial cancer, non-small cell lung cancer, and renal cell carcinoma has been revolutionized in recent years, reporting unprecedented response rates and survival benefits [19,20,21,22,23,24]. Two meaningful examples are the impressive complete response rate of 10% achieved with nivolumab plus ipilimumab combination in metastatic renal cell carcinoma in Checkmate 214 trial and the survival benefits provided by immune checkpoint monoclonal antibodies in metastatic malignant melanoma where—until the approval of ipilimumab in 2011—patients with distant metastases presented 5-year survival rates of approximately 5% [25,26]. Moreover, on the basis of recent results of trials testing immunotherapy alone or in combination with other anticancer agents in different malignancies (e.g., gastric cancer, colorectal cancer, hepatocellular carcinoma, etc.) the number of indications for immune checkpoint monoclonal antibodies is supposed to further increase in the coming years [27,28,29,30,31,32]. However, if the advent of immune checkpoint monoclonal antibodies has certainly been a breakthrough in the therapeutic landscape of a number of hematological and solid tumors, the detection of specific molecular and histological biomarkers predictive of response to immunotherapy remain the current challenge [33]. Thus, the identification of predictive biomarkers for selecting immune checkpoint monoclonal antibodies treatment represents an extremely active area of preclinical and clinical research in medical oncology [34]. This topic is particularly important in malignancies where low response rates to immunotherapy have been observed so far—as in the case of PC.

The measurement of the expression of PD-L1 is considered a biomarker-based strategy able to predict benefit from anti-PD-1 and anti-PD-L1 antibodies in several tumor types [35]. Nonetheless, the predictive value of PD-L1 expression in immune checkpoint monoclonal antibodies varies among different malignancies, and moreover, also PD-L1 negative tumors can respond to immunotherapy [36]; in addition, the positivity cut-off value of PD-L1 and the methods of evaluation of this biomarker show important variations and are not standardized [37]. With regard to PC, it is worth noting that the majority of studies assessing PD-L1 expression concerned primary specimens, and conversely, few data are currently available about secondary lesions [38]. Early reports evidenced that primary PC specimens expressed little or no PD-L1; nonetheless, a number of recent studies have described significant PD-L1 expression in both mCRPC and primary PC (up to 20% and 92%, respectively) [39,40], and interestingly, the expression of PD-L1 does not appear to be a reliable biomarker of response to immune checkpoint monoclonal antibodies in PC [41]. However, methods of PD-L1 evaluation may vary widely in distinct trials and across laboratories, with the presence of different assays and scoring systems to define the cut-off positivity for PD-L1, and thus, the first step before exploring the impact of PD-L1 would most likely be to use a standardized, single method.

A study by Gevensleben et al. evaluated the expression of PD-L1 on primary RP specimens of 209 hormone-treatment-naïve patients, assessing the prognostic impact of PD-L1 expression [42]. According to the results of this study, not only moderate to high PD-L1 expression was significantly associated with higher Ki-67 (*p* < 0.001), androgen receptor expression (*p* < 0.001), and Gleason score (*p* = 0.004), but high PD-L1 patients showed shorter biochemical recurrence-free survival. Thus, this evidence suggested that high PD-L1 expression could represent an independent prognostic factor determining higher risk of recurrence in patients previously affected by PC and that had been subjected to RP.

PD-L1 has also been highlighted as a dynamic biomarker in PC, implicated in mechanisms of resistance to enzalutamide treatment and immune evasion. Firstly, Bishop et al. compared PD-L1/PD-L2 expression in blood dendritic cells (DCs) between mCRPC patients progressing on enzalutamide, treatment-naïve patients and subjects who had responded to enzalutamide treatment [43]. Interestingly, enzalutamide resistance was found to be associated with PD-L1/PD-L2 positivity since patients that progressed to enzalutamide showed more PD-L1/PD-L2 positive DCs than responsive subjects; conversely, lower circulating PD-L1/PD-L2 positive DCs were detected in enzalutamide responders. The study was the first to evidence that changes in PD-L1 expression could represent a mechanism involved in enzalutamide resistance, having the merit to raise the question whether PD-L1 expression could be modulated, and thus representing a viable strategy to enhance the efficacy of immune checkpoint monoclonal antibodies. More recently, a study by Pal and colleagues evaluated circulating levels of immune-suppressive (e.g., GM-CSF, IL-6, IL-10, FGF) and proinflammatory mediators in mCRPC patients receiving enzalutamide or abiraterone [44]. According to the results of this report, subjects resistant to enzalutamide and abiraterone showed high levels of proinflammatory mediators such as IFN-gamma and IL-5; conversely, increased levels of IL-6, IL-10 and FGF were found in responders, further supporting the hypothesis that proinflammatory mediators could be involved in immune evasion and mechanisms of drug resistance.

Similar to PD-L1 expression, patients with MSI-H or mismatch repair deficiency (dMMR) tumors are considered potential candidates for immune checkpoint monoclonal antibodies treatment given the presence of high levels of mutation-associated neoantigens, resulting in genetic hypermutability and higher mutational load [45]. In 2017, the historic approval of the anti-PD-1 agent pembrolizumab by the US Food and Drug Administration (FDA) for unresectable or metastatic MSI-H/dMMR malignancies—regardless of tumor type—that have progressed after prior standard treatment and without satisfactory alternative treatment options, represented the first tissue/site-agnostic approval [46]. In mCRPC, previous reports showed frequencies of MSI-H/dMMR ranging between 1% and 12%, with an overall unclear prevalence [47,48]. As previously indicated by Nava Rodrigues et al. in their integrated analysis of genomic, transcriptomic, and clinical data in two cohorts of PCs, an important discordance may exist among different assays used to identify dMMR tumors [49]. Moreover, the authors highlighted through transcriptomic analysis that dMMR PCs showed higher T-cell infiltration and immune checkpoint-related transcripts, something which supports the use of immune checkpoint monoclonal antibodies in these patients. Finally, four different mutational signatures were identified, with dMMR signatures resulting associated to prominent expression of genes involved in accumulation of myeloid-derived suppressor cells (MDSCs), including VCAM1, NLRP3, and JAK2 [49]. Thus, this report suggested that the efficacy of immunotherapy in dMMR mCRPC could be enhanced through strategies that may cause myeloid cells depletion. More recently, a single institution experience at Memorial Sloan Kettering Cancer Center on 1033 mCRPC patients evidenced that 3.1% of patients (*n* = 32) carried MSI-H/dMMR PC, of whom 21.9% presented a Lynch syndrome-associated germline mutation [50]. In this study, 11 out of 32 patients received an immune checkpoint monoclonal antibody, with six of them achieving >50% reduction in Prostate Specific Antigen (PSA) levels and radiological response in four subjects.

Since PD-L1 and MSI-H/dMMR present important limitations in terms of both sensitivity and specificity, the research of other biomarkers predictive of response to immune checkpoint monoclonal antibodies has evidenced that patients with high somatic mutational load could have higher response rates to immunotherapy [51]. Tumor mutational burden (TMB) represents the most commonly used method to quantify mutational load, often reported as mutations/megabase (mut/Mb) [52]; in other terms, TMB count expresses the number of mutations reported in a megabase of tumor genomic territory [53]. However, although TMB is considered a promising predictive biomarker of response to immunotherapy, there is currently lack of standardization in TMB quantification, something which has generated a lot of debate in terms of the cut-off value determining high TMB across different platforms and cancer types [54,55]. In fact, the number of mut/Mb is extremely variable between and within malignancies, with a wide range from 0.1 mut/Mb to more than 200 mut/Mb [56]; PC is generally considered a tumor with low mutational load, with previous studies reporting around 1 to 2 mut/Mb, values particularly lower compared to those of lung cancer and melanoma [57,58].

Nonetheless, the presence of mutations in the DNA damage repair (DDR) genes (e.g., BRCA1, BRCA2, CDK12, ATM) has been associated with an increased number of DNA errors and high tumor neoantigen expression, and thus could be predictive of response to immune checkpoint monoclonal antibodies [59]. With regard to PC, germline or somatic mutations have been reported in up to 25% of cases of advanced disease, mainly in homologous recombination (HR) repair genes [60]. In fact, high TMB has been observed in patients harboring HR and/or MMR defects, suggesting that immune checkpoint monoclonal antibodies could be an attractive option in these patients [61]. Among DDR mechanisms, the transcription associated Cyclin-Dependent Kinase 12 (CDK12) plays an essential role in DNA damage response and differentiation through a regulation of a number of genes [62]; thus, mutations in CDK12 result in accumulation of DNA damages, carcinogenesis, and the formation of immunogenic neoantigens. Interestingly, around 7% of mCRPCs report biallelic CDK12 mutations [63]. A report by Barrero et al. compared tumor infiltrating lymphocytes (TILs) between 11 PC patients with biallelic CDK12 mutations and 47 PC patients with monoallelic CDK12 mutations or no mutations [64]. According to the results of this study, TILs count was higher in biallelic CDK12 altered PCs, with higher CD8+ (95% CI 0.24–2.18; *p* = 0.02) and higher CD4+ (95% CI 0.11–1.94; *p* = 0.03) compared to patients harboring other mutations and/or monoallelic CDK12 mutations. Another study by Petitprez et al. suggested that PD-L1 and CD8+ TILs in node-positive PC patients were associated with higher risk of disease progression [65].

Recent reports have been focused as well on the relationship between ductal histology—a highly aggressive and rare histological type—and DDR germline mutations. Firstly, a study by Guedes et al. on 150 PC patients evidenced that intraductal/ductal histology was more common in patients with DDR germline mutations compared to germline-negative subjects [66]. In this report, MSH2 loss was associated with hypermutation and higher TILs density, also resulting correlated with high-grade tumors with primary Gleason pattern 5. Although based on a small sample size, a study by Schweizer et al. detected that four out of 10 patients with ductal PC were dMMR, and 3 of them were also MSI-H [67]. In particular, one of these MSI-H/dMMR patients experienced an outstanding response to pembrolizumab, with a dramatic decline of PSA value during immune checkpoint monoclonal antibodies treatment.

Lastly, since the subgroup of AR-V7 positive patients has been associated with DDR genes mutations [68] and nivolumab plus ipilimumab showed encouraging efficacy and a manageable safety profile in AR-V7 positive, DDR mutated mCRPCs, this subset is considered particularly promising in terms of response to immune checkpoint monoclonal antibodies [69]. More specifically, a phase II biomarker-driven trial suggested that AR-V7 positive patients with DDR mutations (BRCA2, ATM, MSH6, FANCM, FANCA, and POLH) treated with nivolumab plus ipilimumab presented a statistically significant benefit in terms of PFS (lack of progression ≥24 weeks) compared to AR-V7 positive, DDR negative subjects [69].

Taken together, all these data suggest that some subgroups of PC patients could benefit from immunotherapy [70]. These could include subjects with aggressive tumors (e.g., Gleason pattern 5, ductal histology, etc.), PCs harboring homologous recombination deficiency (HRD) mutations, the AR-V7 positive subgroup, and patients with biomarker of response to immune checkpoint monoclonal antibodies such as high TMB and dMMR [71]. Nonetheless, the overall modest activity of immune checkpoint monoclonal antibodies in PC deserves well-designed, tailor-made trials as well as better biomarkers to improve the predictive capacity to unveil responders. In fact, rather than a single biomarker, an approach based on the integration of different biomarkers could most likely help in improving the understanding of the role of immune checkpoint monoclonal antibodies in malignancies where immunotherapy has reported low ORRs, such as mCRPC.

## 3. Prostate Cancer Vaccines

Cancer vaccines are able to prime the immune system to recognize tumor-associated antigens, thus eliciting T cell response [72]. With regard to PC, the expression of several types of tumoral antigens (including PSA, PAP, PSMA, etc.) have provided the rationale for a number of investigations assessing these agents [73,74] and the only cancer vaccine approved by the US FDA for mCRPC is sipuleucel-T—so far. Sipuleucel-T is an autologous cellular immunotherapy generated by the extraction, incubation with PA2014—a recombinant fusion protein of GM-CSF and PAP—and subsequent re-infusion of activated antigen-presenting cells (APCs), eliciting an antitumor immune response (Figure 1) [75]. This process, including leukapheresis, cellular activation and reinfusion, is repeated every two weeks for a total of three doses. In 2010, the pivotal IMPACT trial conducted by Kantoff and colleagues evidenced an increased overall survival (OS) in patients treated with sipuleucel-T [76]. In this phase III study, 512 patients were randomly allocated to receive sipuleucel-T or placebo, with the cancer vaccine conferring a statistically significant OS advantage (25.8 months versus 21.7 months; HR, 0.78; 95% CI, 0.61–0.98; *p* = 0.003); additionally, sipuleucel-T was well tolerated, with most common adverse events including fever and flu-like symptoms. However, it is worth noting that no differences were detected in terms of progression-free survival (PFS) or PSA decline, since less than 3% of sipuleucel-T patients showed a PSA decline of 50% or greater. Moreover, a subsequent analysis of the IMPACT trial which stratified enrolled patients by PSA levels, suggested that sipuleucel-T could be more effective in patients with low tumor burden [77]. In fact, the difference in OS was only 2.8 months between sipuleucel-T arm and placebo arm in patients in the highest quartile of PSA; conversely, according to this analysis, a survival benefit of 13 months was detected in the lowest quartile of PSA. Sipuleucel-T has been the first cellular therapeutic vaccine approved by the FDA, thus representing an historical step for cancer vaccines in medical oncology; nonetheless, more recent trials tempered the enthusiasm for this agent and, as suggested by a large real-world study conducted by Caram and colleagues on 7272 mCRPC patients, only one out of 10 cases were treated with sipuleucel-T, indicating the overall limited use of this cancer vaccine in everyday clinical practice [78,79,80].

Nonetheless, the documented survival benefit associated with sipuleucel-T in the IMPACT trial has paved the way towards a number of trials assessing the efficacy and safety of cancer vaccines, used as monotherapy or in combination with other drugs with potential anticancer activity [81]. Among the agents tested in mCRPC, it is worth mentioning PROSTVAC-VF, GVAX, and DCVAC/PCa.

The cancer vaccine PROSTVAC-VF is a virus-based vaccine targeting PSA. In fact, PROSTVAC-VF consists of a vaccinia virus-based prime acting as a primary vaccination, followed by multiple boosts which employ a recombinant fowlpox vector expressing PSA [82]. Interestingly, PROSTVAC-VF includes three elements able to elicit a robust anti-PSA immune response and which constitutes the TRICOM, the TRIad of CO-stimulatory Molecules: CD-80 (B7.1), which binds to CD28 and plays a role in T cells activation, and the two adhesion molecules ICAM-1 (Intercellular Adhesion Molecule 1) and LFA-3 (Lymphocyte Function-associated Antigen 3), that strengthen the interaction between T cells and APCs (Figure 2) [83]. Despite the promising data of a phase II study comparing PROSTVAC-VF versus placebo in 122 mCRPC patients highlighted a survival advantage in PROSTVAC arm, the more recent PROSPECT trial did not confirm this benefit [84,85]. In fact, this randomized phase III trial enrolling 1297 patients to PROSTVAC-VF plus GM-CSF, to PROSTVAC-VF plus GM-CSF placebo or to double placebo, was discontinued early on. Nonetheless, since antiandrogens and cytotoxic chemotherapy are able to induce an immunogenic modulation, there are currently ongoing trials aimed to ascertain the role of PROSTVAC-VF combined with immune checkpoint monoclonal antibodies, docetaxel, or nonsteroidal antiandrogens (NCT01867333, NTC01875250, NCT02933255, NCT02506114, NCT02649855).

Another cancer vaccine, GVAX, consists in genetically modified tumor cells able to secrete GM-CSF, with a mechanism of action based on the use of two prostate cell lines, LN-CaP (hormone-sensitive cancer cell lines deriving from nodal site of metastasis) and PC-3 (hormone-refractory cancer cell lines deriving from bone metastases) [86]. Despite early evidence supported the use of GVAX in hormone-naïve PCs, the phase III VITAL-1 and VITAL-2 trials showed disappointing results [87,88]. More specifically, the randomized, open-label VITAL-2 trial comparing GVAX plus docetaxel versus docetaxel alone in taxane-naïve mCRPC, was closed prematurely given an increased mortality rate in the experimental arm. Similarly, the VITAL-1 study comparing GVAX versus docetaxel and prednisone was terminated after an early futility analysis.

The DCVAC/PCa, a more recent cancer vaccine composed of mature DCs and LNCaP, has been evaluated in combination with docetaxel chemotherapy in a phase I/II trial [89,90]. In this study, including 25 mCRPC patients, the combination of DCVAC plus chemotherapy showed a manageable safety profile, with a median OS of 19.0 months. On the basis of these results, the randomized phase III VIABLE trial has assessed DCVAC/PCa plus docetaxel chemotherapy. The results of this trial are not available so far.

Another vaccine strategy involves the use of attenuated vaccines derived from Listeria Monocytogenes (LM), the causative agent of listeriosis which induces a relevant immune response after the entrance into APCs through phagocytosis (Figure 3) [91]. The rationale of LM vaccines relies on the stimulation of immune response avoiding the pathogenic features of this bacterium, using LM-Listeriolysin O (LLO) therapies [92]. Among these therapies, the listeria attenuated vaccine ADXS31-142 targeting PSA has shown promising results in terms of antitumor efficacy in PC preclinical murine models since previous reports regarding the combination of ADXS31-142 and anti-PD-1 agents have resulted in prolonged survival in treated mice [93,94]. An ongoing phase I trial is currently testing the role of ADXS31-142 as single-agent and combined with pembrolizumab in mCRPC patients (NCT02325557); preliminary results of this study have highlighted that the 14% of patients treated with single-agent vaccine and the 43% of subjects receiving combination therapy have shown a decreased PSA post-baseline [95].

## 4. Single Agent Immune Checkpoint Inhibitors Trials

Different monoclonal antibodies against PD-1, PD-L1, or CTLA-4 have been tested in the treatment of mCRPC with mostly disappointing results.

A first pivotal trial evaluated single 3 mg/kg dose of Ipilimumab, a fully human anti-CTLA-4 IgG monoclonal antibody, in 14 patients with hormone refractory PC [96]. Two patients showed PSA declines of ≥50% and eight patients had a PSA decline <50%. Using the Response Evaluation Criteria in Solid Tumors (RECIST), none of the two patients with measurable disease at baseline and repeated scans showed an objective response. Immune adverse events were limited to a single patient (grade 3 rash and pruritus).

A phase III trial randomized patients with at least one bone metastasis from CRPC progressed after docetaxel to receive bone-directed radiotherapy (8 Gy in one fraction) followed by either ipilimumab 10 mg/kg or placebo every 3 weeks for up to four doses (CA184-043 trial) [97]. The experimental treatment with ipilimumab did not increase OS, the primary endpoint (median OS 11.2 months with ipilimumab versus 10.0 months with placebo; HR 0.85, 95% CI 0.72–1.00, *p* = 0.053). Nonetheless, ipilimumab increased median PFS (4.0 months with ipilimumab versus 3.1 months with placebo; HR 0.70, 95% CI 0.61–0.82; *p* < 0.0001) and prespecified subgroup analyses showed that the benefit of ipilimumab was higher for patients with favorable prognostic factors, especially the presence or absence of visceral metastases. In terms of safety, grade 3–4 adverse events occurred in the 26% (*n* = 101) of patients in the ipilimumab arm and the 3% (*n* = 11) of the placebo group, with diarrhea representing the most common event (16% and 2% of patients in the ipilimumab arm and the placebo arm, respectively). Four toxic deaths were reported in the ipilimumab group (1% of patients).

A subsequent randomized phase III trial (CA184-095) evaluated the safety and efficacy of ipilimumab versus placebo in the first-line treatment of patients with asymptomatic or minimally symptomatic mCRPC without visceral metastases [98]. Ipilimumab did not meet its primary endpoint of OS (median OS 28.7 months in the ipilimumab arm versus 29.7 months in the placebo arm; HR 1.11; 95.87% CI, 0.88 to 1.39; *p* = 0.3667). Some antitumor activity was shown in terms of median PFS (5.6 months in the ipilimumab arm versus 3.8 with placebo arm; HR 0.67; 95.87% CI, 0.55 to 0.81) and PSA response rate (23% with ipilimumab versus 8% with placebo). According to this phase III trial, diarrhea was the only grade 3–4 immune-related adverse event (irAEs) reported in the 15% of enrolled subjects receiving ipilimumab. Importantly, irAEs were observed in 31% of subjects in the ipilimumab arm and the 2% of patients treated with ipilimumab (*n* = 9) died due to irAEs.

The monoclonal antibody against PD-1 pembrolizumab has been tested in mCRPC pretreated patients. The phase Ib KEYNOTE-028 study enrolled patients with PD-L1 ≥1% of tumor or stromal cells and patients received pembrolizumab 10 mg/kg every two weeks until disease progression or intolerable toxicity for up to 24 months [99]. Of the 23 patients enrolled, 73.9% received at least two prior therapies for metastatic disease. Pembrolizumab achieved an overall response rate (ORR) of 17.4% with four partial responses and eight stable disease. Median PFS and OS were 3.5 and 7.9 months, respectively. The 60.9% of patients presented irAEs, the most frequent of which was nausea (13%); grade 3–4 irAEs (peripheral neuropathy, lipase increase, fatigue, and asthenia) were reported in the 17.3% of patients (*n* = 4). Lastly, no treatment-related deaths or discontinuation occurred during this trial.

In the phase II trial KEYNOTE-199 patients pretreated with docetaxel and one or more targeted endocrine therapies for mCRPC received pembrolizumab 200 mg every three weeks for up to 35 cycles [100,101]. mCRPC patients were enrolled in three cohorts: cohorts 1 (133 patients) and 2 (66 patients) enrolled patients with RECIST-measurable disease PD-L1-positive and PD-L1-negative, respectively, while cohort 3 (59 patients) enrolled patients with bone-predominant disease, regardless of PD-L1 expression. ORR, the primary endpoint, was 6% in cohort 1 and 3% in cohort 2 [101]. Treatment-related adverse events (TRAEs) of any grade/grade 3–5 occurred in 57%/16% in cohort 1, 60%/15% in cohort 2, and 71%/17% in cohort 3. Median OS was 9.5 months in cohort 1, 7.9 months in cohort 2, and 14.1 months in cohort 3 [100]. These results are encouraging for pretreated patients with limited treatment options at disposal, with about 25% of patients having received both enzalutamide and abiraterone. Pembrolizumab activity was demonstrated in both PD-L1 positive and negative patients and bone-predominant or RECIST-measurable disease. Exploratory biomarker analysis did not identify a correlation between response to pembrolizumab and DDR genes. With regard to toxicity, the 60% of patients presented irAEs, in the 15% of cases of grades 3, 4 or 5. Finally, the 5% of patients discontinued treatment due to pembrolizumab-related adverse events.

The monoclonal antibody against PD-L1 atezolizumab has been tested in a phase Ia (PCD4989g; NCT01375842) in patients with mCRPC previously treated enzalutamide and/or sipuleucel-T [102]. Atezolizumab was administered at the dose of 1200 mg IV every three weeks. In the 15 patients evaluated, the landmark 12-months OS rate was 55.6% and the six-months PFS rate was 26.7%. One patient (9%) achieved a partial response per immune-related response criteria (irRC) and five patients (45%) had SD per RECIST version 1.1 and irRC. Nine patients (60%) presented irAEs with one grade 3 hyponatremia and no grade 4–5 toxicities.

Avelumab, a human IgG1 monoclonal antibody that binds to PD-L1, was investigated in an open-label phase Ia, dose-escalation trial (part of the JAVELIN Solid Tumor trial) [103]. Avelumab was administered as one-hour intravenous infusion every two weeks at four different doses (1 mg/kg, 3 mg/kg, 10 mg/kg, and 20 mg/kg) with dose-level cohort expansions. The dose of 10 mg/kg every two weeks was chosen for further development.

The expansion cohort of the JAVELIN Solid Tumor trial evaluated avelumab 10 mg/kg in 18 patients with mCRPC progressed on previous treatments [104]. The experimental treatment was generally well tolerated with 15 patients experiencing grade ≤2 and two patients grade 3 asymptomatic irAEs (amylase and lipase elevations). In terms of response, seven patients achieved a stable disease >24 weeks post treatment, and six patients had progressive disease after first restaging scans at six weeks. (PSADT) prior to avelumab was compared with PSADT after three months (m) of treatment. Three of the 17 evaluable patients had a prolonged PSA doubling time (PSADT), seven had stable PSADT and seven decreased PSADT.

## 5. Combination Immune Checkpoint Inhibitors Trials

In order to amplify treatment activity and improve patients’ outcomes, multiple combination strategies have been investigated. One of the main strategies experimented in multiple types of cancer is the association of an anti-PD-1 and an anti-CTLA-4. The CheckMate 650 trial is a phase II study investigating the association of nivolumab, an anti-PD-1, with ipilimumab in asymptomatic or minimally symptomatic mCRPC patients progressed to novel androgen receptor targeted agents and that did not receive chemotherapy for mCRPC (cohort 1) and patients progressed to taxane-based chemotherapy (cohort 2) [105]. Patients received nivolumab 1 mg/kg plus ipilimumab 3 mg/kg every three weeks for four doses, then nivolumab 480 mg every four weeks. The preplanned interim efficacy/safety analysis showed an ORR of 26% and 10% in cohorts 1 and 2. Interestingly, in both cohorts ORR was higher in patients with PD-L1 ≥ 1%, DDR, HRD, or above-median TMB (with a cut-off between above/below median of 74.5 mutations/pt). Regarding safety analysis, 39% and 51% of patients in cohorts 1 and 2 had grade 3–4 irAEs while one grade 5 event was reported in each cohort.

The combination of nivolumab plus ipilimumab has been investigated in the subgroup of patients with AR-V7 positive mCRPC patients as well [106]. Fifteen patients were enrolled in this phase II study (NCT02601014), 60% of patients had received ≥4 prior regimens and 40% harbored somatic and/or germline mutations in DDR genes. In fact, as already mentioned, AR-V7 positive tumors have been associated with a higher presence of DDR gene alterations and higher mutation load [107]. The combination approach showed encouraging results in DDR gene altered patients but not in the overall population in terms of PSA responses (33% versus 0%; *p* = 0.14), ORR (40% versus 0%; *p* = 0.46), PSA-PFS (HR 0.19; *p* < 0.01), PFS (HR 0.31; *p* = 0.01,), OS (HR 0.41; *p* = 0.11).

As already mentioned, one of the mechanisms involved in enzalutamide resistance could be represented by changes in PD-L1 expression considering that patients progressing to enzalutamide resulted to be associated with a higher expression of PD-L1/PD-L2 positive DCs [53]. Thus, one of the strategies to overcome this type of resistance exploiting this immunomodulatory effect is to combine immune checkpoint monoclonal antibodies and androgen receptor targeted agents. A phase II trial investigated the combination of pembrolizumab plus enzalutamide in patients progressed to enzalutamide monotherapy that could have been previously treated with sipuleucel-T and abiraterone or chemotherapy for castration-sensitive disease [108]. This study suggested that pembrolizumab added to enzalutamide has activity in this population of patients with mCRPC. In particular, five (18%) patients obtained a PSA decline of ≥50% with two long responders and three (25%) patients achieved an objective response. Of these, one had MSI high disease.

Furthermore, the phase II trial KEYNOTE-199 included two cohorts that investigated the combination of pembrolizumab and enzalutamide in chemotherapy-naïve patients previously treated with enzalutamide (cohort 4 RECIST measurable disease, cohort 5 bone-predominant disease) [109]. Pembrolizumab was administered at the dose of 200 mg every three weeks with enzalutamide for up to 35 cycles or until progression or intolerable toxicity. Combination treatment resulted in modest antitumor activity with an ORR of 12% in cohort 4 and disease control rate of 51% in cohort 4 and 51% in cohort 5. The safety profile resulted to be manageable with any grade TRAEs occurring in 75% of patients in cohort 4 and 69% in cohort 5 while grade 3–5 TRAEs were reported in 26% of cohort 4 patients and 24% of cohort 5 patients. The finding that a small percentage of patients respond to immune checkpoint monoclonal antibodies but responders present durable responses underline the need for predictive biomarkers to immunotherapy.

At the recent ASCO Genitourinary Cancers Symposium 2020 the results of the phase Ib/II KEYNOTE 365 trial were presented. This study enrolled patients with mCRPC in three different cohorts: cohort A (84 patients) investigated the combination of pembrolizumab plus the poly ADP ribose polymerase inhibitor (PARPi) olaparib in molecularly unselected patients previously treated with docetaxel and second-generation hormone therapy; cohort B (104 patients) enrolled patients previously treated with abiraterone or enzalutamide to receive pembrolizumab plus docetaxel and prednisone; cohort C (102 patients) explored the combination of pembrolizumab plus enzalutamide in abiraterone-pretreated patients. In cohort A, PSA response rate was 9%, ORR was 8% (2/24 patients with RECIST measurable disease) with two partial responses, disease control rate was 22% [110]. Median radiographic PFS was 4.3 months and median OS was 14 months. The safety profile was consistent with individual profile of each agent and grade 3–5 TRAEs occurred in 35% patients. In cohort B, PSA response rate was 28%, ORR was 18% (7/39 patients with RECIST measurable disease) with seven partial responses, disease control rate was 51% [111]. Median radiographic PFS was 8.3 months and median OS 20.4 months. TRAEs occurred in 100 patients (96%) with 40% of patients experiencing grade 3–5 TRAEs and five deaths for adverse events. In cohort C, PSA response rate was 22%, ORR was 12% (3/25 patients with RECIST measurable disease) with two complete responses and one partial response, disease control rate was 35% [112]. Median radiographic PFS was 6.1 months and median OS 20.4 months. TRAEs were reported in 90% with grade 3–5 in 39% of patients and three deaths due to adverse events. Considering these promising results, the combination of pembrolizumab and enzalutamide is currently being evaluated in a randomized phase III trial versus enzalutamide plus placebo (KEYNOTE 641, NCT03834493).

The anti-PD-L1 atezolizumab (1200 mg every three weeks) has been investigated in combination with enzalutamide in the randomized phase III IMbassador250 trial versus enzalutamide alone in 759 patients with mCRPC progressed to abiraterone and docetaxel or ineligible for taxane-based therapy [113]. The experimental treatment failed to improve OS, the primary endpoint, or radiographic PFS, a secondary endpoint. The 12-month OS rates were 60.6% in the combination arm and 64.7% in the enzalutamide arm. Median OS was 15.2 months versus 16.6 months (HR 1.12, 95% CI 0.91–1.37, *p* = 0.28) and median radiographic PFS was 4.2 months versus 4.1 months (HR 0.90, 95% CI 0.75–1.07) in the experimental arm and in the control arm, respectively. No subgroup appeared to benefit from the combination treatment. In terms of safety profile, TRAEs have been observed in the 77.8% and the 51.1% of patients receiving atezolizumab plus enzalutamide or enzalutamide monotherapy, respectively. Moreover, grade 3–4 TRAEs have been reported in 28.3% of patients treated with the immune-based combination and the 9.6% of enzalutamide arm. Lastly, grade 5 TRAEs occurred in 1.9% and 0.3% of enrolled subjects.

Lastly, durvalumab, a human IgG1 monoclonal antibody against PD-L1, combined with olaparib has been investigated in a phase II trial (NCT02484404) in 17 mCRPC patients progressed to enzalutamide and/or abiraterone with or without DDR mutations [114]. Median radiographic PFS was 16.1 months in all patients and for those with alterations in DDR genes with a 12-month rPFS of 51.5%. Four responders harbored germline alterations in DDR genes. The combination treatment resulted to have acceptable toxicity.

Considering the promising but still limited results of immunotherapy in prostate cancer many experimental approaches are being evaluated to overcome resistance mechanisms. Ongoing clinical trials investigating different immunotherapy approaches, either alone or in combination, are reported in Table 1.

## 6. Conclusions

Overall, it is still unclear whether immunotherapy could provide clinical benefits in nmCRPC patients, with several studies reporting disappointing results so far. Nonetheless, the exploration of potential biomarkers able to identify responders is warranted, with immune-based combinations with cytotoxic chemotherapy or other anticancer agents having the potential to increase response rates in subgroups of nmCRPC patients. Additional results from a number of ongoing trials are expected soon, with an aim to define whether there could finally be a role—and what kind of role—for immunotherapy in this malignancy.

## Figures and Tables

**Figure 1 cells-09-02051-f001:**
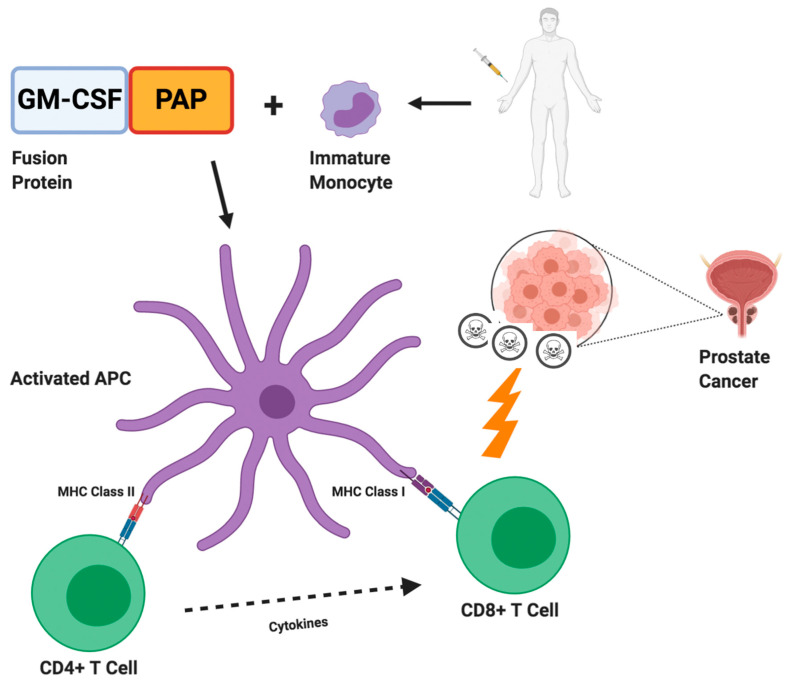
Sipuleucel-T in prostate cancer. Firstly, the cancer vaccine sipuleucel-T requires leukapheresis of immature immune cells; leukapheresis is then followed by incubation with specific fusion protein (PA2024), consisting of prostatic acid phosphatase (PAP) coupled with granulocyte-macrophage colony stimulating factor (GM-CSF). Subsequently, cells are re-infused allowing for APC maturation and activation of CD4+ and CD8+ T cells, which in turn are able to recognize and kill PAP presenting tumor cells.

**Figure 2 cells-09-02051-f002:**
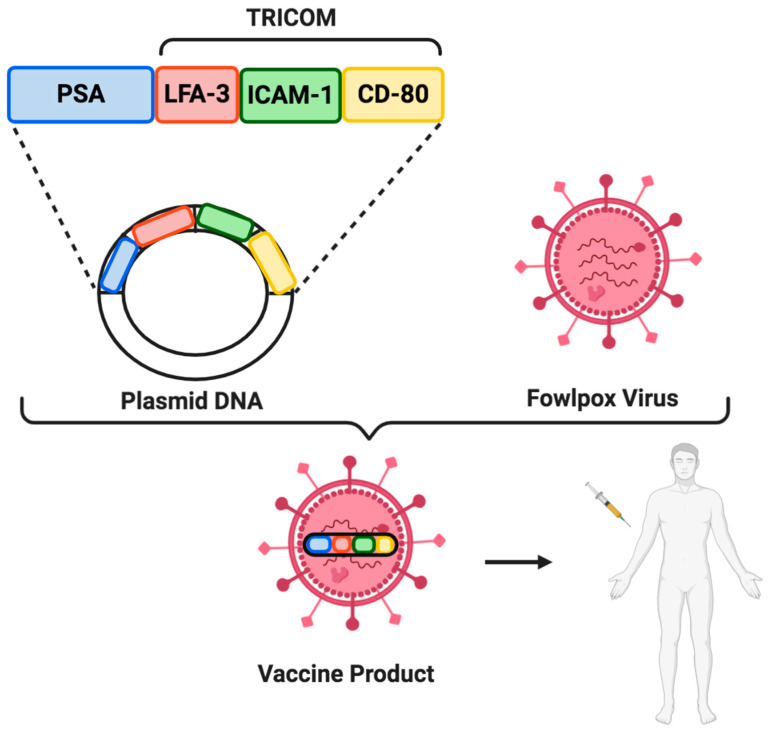
PROSTVAC-VF vaccine. PROSTVAC-VF consists of a recombinant vaccinia vector followed by multiple booster vaccination using a recombinant fowlpox vector. Both vectors contain PSA and the TRIad of CO-stimulatory Molecules (TRICOM), which in turn includes B7-1, ICAM-1, and LFA-3.

**Figure 3 cells-09-02051-f003:**
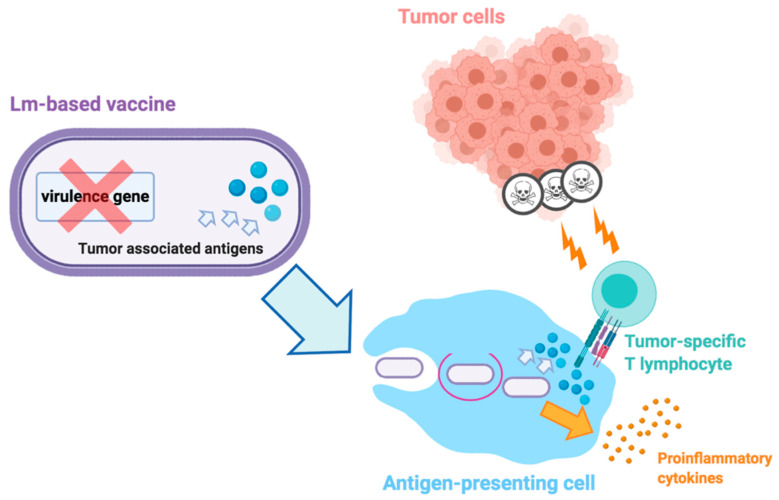
Simplified mechanisms of action of Listeria Monocytogenes (Lm)-based vaccine in cancer. Firstly, Lm-based vectors are attenuated, with removal of one or more virulence genes. When attenuated, these vaccines are particularly rich in tumor-associated antigens; following administration, Lm-based vaccines infect antigen-presenting cells (APCs), escape phagocytosis and secrete tumor-associated antigens; these antigens are involved in the secretion of proinflammatory cytokines, the upregulation of costimulatory molecules, and the activation of tumor-specific cytolytic T lymphocytes which produce antitumor responses and cell death.

**Table 1 cells-09-02051-t001:** Ongoing clinical trials of immunotherapy in prostate cancer.

NCT (clinicaltrials.gov)	Phase	Setting	Number of Patients	Experimental Arm	Control Arm	Mechanism of Action	Status
NCT03170960	I/II	mCRPC	1732	Atezolizumab + Cabozantinib	\	Atezolizumab: anti-PD-L1 Cabozantinib: tyrosine kinase inhibitor	Recruiting
NCT03673787	I/II	mCRPC (PTEN loss)	51	Atezolizumab + Ipatasertib	\	Atezolizumab: anti-PD-L1 Ipatasertib: inhibitor of the serine/threonine protein kinase Akt	Active, not recruiting
NCT03024216	I	mCRPC	37	Atezolizumab + Sipuleucel-T	\	Atezolizumab: anti-PD-L1 Sipuleucel-T: autologous cellular immunotherapy	Active, not recruiting
NCT02655822	I	mCRPC	336	Atezolizumab + Ciforadenant	\	Atezolizumab: anti-PD-L1 Ciforadenant: inhibitor of the adenosine A2A receptor	Recruiting
NCT02788773	II	mCRPC	52	Durvalumab +/− Tremelimumab	\	Durvalumab: anti-PD-L1 Tremelimumab: anti-CTLA-4	Active, not recruiting
NCT03204812	II	mCRPC	27	Durvalumab + Tremelimumab	\	Durvalumab: anti-PD-L1 Tremelimumab: anti-CTLA-4	Active, not recruiting
NCT02643303	I/II	mCRPC	102	Durvalumab + Tremelimumab + PolylCLC	\	Durvalumab: anti-PD-L1 Tremelimumab: anti-CTLA-4 PolylCLC: tool-like receptor agonist	Recruiting
NCT03385655	II	mCRPC	500	Durvalumab + Tremelimumab	Carboplatin or Ipatasertib or Savolitinib or Darolutamide or Adavosertib or CFI-400945	Durvalumab: anti-PD-L1 Tremelimumab: anti-CTLA-4 Carboplatin: platinum salt Ipatasertib: AKT inhibitor Savolitinib: cMET inhibitor Darolutamide: non-steroidal androgen receptor antagonist Adavosertib: WEE-1 inhibitor CFI400945: PLK4 inhibitor	Recruiting
NCT02740985	I	mCRPC	307	Durvalumab + AZD4635	\	Durvalumab: anti-PD-L1 AZD4635: antagonist of the adenosine A2A receptor	Recruiting
NCT03330405	II	mCRPC	214	Avelumab + Talazoparib	\	Avelumab: anti-PD-L1 Talazoparib: PARP inhibitor	Active, not recruiting
NCT03409458	I/II	mCRPC	52	Avelumab + PT-112	\	Avelumab: anti-PD-L1 PT-112: A platinum agent complexed to a pyrophosphate ligand	Recruiting
NCT02933255	I/II	Cohort 1: mCRPC Cohort 2: localized prostate cancer	29	Nivolumab +/− PROSTVAC-V/F	\	Nivolumab: anti-PD-1 PROSTVAC-V/F: A vaccine composed of rilimogene galvacirepvec (a recombinant vaccinia virus) and rilimogene glafolivec (a recombinant fowlpox virus)	Recruiting
NCT03600350	II	nmHSPC	41	Nivolumab + pTVG-HP	\	Nivolumab: anti-PD-1 pTVG-HP: A vaccine containing plasmid DNA encoding human prostatic acid phosphatase	Recruiting
NCT03572478	I/II	mCRPC	12	Nivolumab + Rucaparib	\	Nivolumab: anti-PD-1 Rucaparib: PARP inhibitor	Active, not recruiting
NCT03040791	II	mCRPC (DDR defects)	29	Nivolumab	\	Nivolumab: anti-PD-1	Recruiting
NCT03338790	II	mCRPC	330	Nivolumab + Rucaparib or Docetaxel or Enzalutamide	\	Nivolumab: anti-PD-1 Rucaparib: PARP inhibitor Docetaxel: taxane Enzalutamide: androgen receptor targeted agent	Recruiting
NCT03061539	II	mCRPC (dMMR, DDR defects, high TILs)	175	Nivolumab + Ipilimumab	\	Nivolumab: anti-PD-1 Ipilimumab: anti-CTLA-4	Recruiting
NCT03570619	II	mCRPC (biallelic CDK12 loss)	40	Nivolumab + Ipilimumab	\	Nivolumab: anti-PD-1 Ipilimumab: anti-CTLA-4	Recruiting
NCT03532317	I	mCRPC	20	Neoantigen DNA Vaccine + Nivolumab/Ipilimumab + PROSTVAC-V/F	\	Nivolumab: anti-PD-1 Ipilimumab: anti-PD-L1 PROSTVAC-V/F: A vaccine composed of rilimogene galvacirepvec (a recombinant vaccinia virus) and rilimogene glafolivec (a recombinant fowlpox virus)	Recruiting
NCT02985957	II	mCRPC	497	Nivolumab/Ipilimumab or Ipilimumab	Cabazitaxel	Nivolumab: anti-PD-1 Ipilimumab: anti-PD-L1 Cabazitaxel: microtubule inhibitor	Recruiting
NCT04109729	IB/II	mCRPC	36	Nivolumab + Radium-223	\	Nivolumab: anti-PD-1 Radium-223: alpha particle-emitting radiotherapy drug	Recruiting
NCT01688492	I/II	mCRPC	57	Ipilimumab + Abiraterone Acetate + prednisone	\	Ipilimumab: anti-CTLA-4 Abiraterone Acetate: androgen receptor targeted agent	Active, not recruiting
NCT03093428	II	mCRPC	45	Pembrolizumab + Radium-223	\	Pembrolizumab: anti-PD-1 Radium-223: alpha particle-emitting radiotherapy drug	Active, not recruiting
NCT03506997	II	mCRPC	100	Pembrolizumab	\	Pembrolizumab: anti-PD-1	Recruiting
NCT03248570	II	mCRPC (with or without DDR defects)	50	Pembrolizumab	\	Pembrolizumab: anti-PD-1	Recruiting
NCT02499835	I/II	mCRPC	72	Pembrolizumab + pTVG-HP	\	Pembrolizumab: anti-PD-1 pTVG-HP: A vaccine containing plasmid DNA encoding human prostatic acid phosphatase	Recruiting
NCT02998567	I	mCRPC	34	Pembrolizumab + guadecitabine	\	Pembrolizumab: anti-PD-1 Guadecitabine: dinucleotide antimetabolite of a decitabine linked via phosphodiester bond to a guanosine	Active, not recruiting
NCT03406858	II	mCRPC	33	Pembrolizumab + HER2Biarmed activated T-cells	\	Pembrolizumab: anti-PD-1 HER2Biarmed activated T-cells	Recruiting
NCT03834493	III	mCRPC	1200	Pembrolizumab + Enzalutamide	Enzalutamide + Placebo	Pembrolizumab: anti-PD-1 Enzalutamide: androgen receptor targeted agent	Recruiting
NCT03473925	II	mCRPC	120	Pembrolizumab + Navarixin	\	Pembrolizumab: anti-PD-1 Navarixin: antagonist of CXCR1 and CXCR2	Active, not recruiting
NCT03007732	II	mHSPC	42	Pembrolizumab + SBRT +/− SD-101	\	Pembrolizumab: anti-PD-1 SD-101: synthetic CpG oligonucleotide that stimulates Toll-like receptor 9	Recruiting

mCRPC: metastatic castration resistant prostate cancer; nmHSPC: non-metastatic hormone sensitive prostate cancer; mHSPC: metastatic hormone sensitive prostate cancer; SBRT: stereotactic body radiation therapy; DDR: DNA-damage repair; dMMR: deficient mismatch repair; TILs: tumor infiltrating lymphocytes; ORR: overall response rate; PARP: poly ADP ribose polymerase.

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
