# Peer review of "Is There a Role for Immunotherapy in Prostate Cancer?"

_cells, 2020, doi:10.3390/cells9092051_

Round 1

Reviewer 1 Report

Rizzo et al., reviewed the current knowledge on immunotherapies in metastatic castration-resistant prostate cancer (mCRPC), particularly focusing on cancer vaccine and ICI therapies. They mentioned early reports suggesting that the PD-L1 levels in primary PC were 92%, different from those in mCRPC, which were ~20%. They introduced case studies of clinical trials for patients with mCRPC. Their summary revealed that ICI monotherapies are not so effective to mCRPC patients, but combinational therapies are hopeful in an increase in objective response rate (ORR). Tumor mutation burden (TMB) consisting of many parameters appeared important in encouraging results of immunotherapies. Provenge, the only vaccine approved by the US FDA for mCRPC, increased overall survival (OS) but failed to detect significance on progression-free survival or PSA decline. Development of additional vaccine therapies is going on.

Concerns:

  1. This review nicely summarized the current clinical studies on mCRPC to contribute to facilitating the next design of immunotherapy in mCRPC. Abstract should be polished to clarify the points in this review.

  1. There are many abbreviations in the text, which disturbs the easy understanding of the content. Furthermore, the definition of the abbreviations is sometimes insufficient. Please provide the abbreviation list.

Author Response

Response to reviewer #1

Dear Reviewer, thank you for the time spent to revise our work. We have modified the paper in response to your comments. We hope the revised manuscript will better suit Cells.

#1: This review nicely summarized the current clinical studies on mCRPC to contribute to facilitating the next design of immunotherapy in mCRPC. Abstract should be polished to clarify the points in this review.

#1: Thank you for pointing this out. We modified part of the Abstract, according to your comments. In fact, we added two sentences which could help to better define the rationale of our work, with a view to suggest possible directions of future research.

In particular, we modified as follows (red):

Moreover, despite sipuleucel-T represents the only cancer vaccine approved by Food and Drug Administration (FDA) for mCRPC following the results of the IMPACT trial, the use of this agent is relatively limited in everyday clinical practice.

(…)

In this review, we aim at underlining the failures and promises of immunotherapy in prostate cancer, summarizing current state of art regarding cancer vaccines and ICIs and discussing future research directions in this immunologically “cold” malignancy. 

#2: There are many abbreviations in the text, which disturbs the easy understanding of the content. Furthermore, the definition of the abbreviations is sometimes insufficient. Please provide the abbreviation list.

#2: Thank you for your suggestion. We added the abbreviation list (in alphabetical order) at the end of the manuscript, as follows (green):

Abbreviation list

APCs: antigen-presenting cells

CDK12: Cyclin-Dependent Kinase 12

CTLA-4: cytotoxic T-lymphocyte antigen-4

DCs: dendritic cells

DDR: DNA damage repair

GM-CSF: granulocyte-macrophage colony stimulating factor

dMMR: mismatch repair deficiency

HR: homologous recombination

HRD: homologous recombination deficiency

ICAM-1: intercellular adhesion molecule 1

ICIs: immune checkpoint inhibitors

LFA-3: lymphocyte function-associated antigen 3

LLO: listeriolysin O

LM: listeria monocytogenes

mCRPC: metastatic castration-resistant prostate cancer

MDSCs: myeloid-derived suppressor cells

mHSPC: metastatic hormone sensitive prostate cancer

MSI-H: high microsatellite instability

Mut/Mb: mutations/megabase

nmHSPC: non-metastatic hormone sensitive prostate cancer

ORR: overall response rate

OS: overall survival

PAP: prostatic acid phosphatase

PARPi: poly ADP ribose polymerase inhibitor

PC: prostate cancer

PD-1: programmed death receptor 1

PFS: progression-free survival

PSA: Prostate Specific Antigen

PSADT: PSA doubling time

RECIST: Response Evaluation Criteria in Solid Tumors

RP: radical prostatectomy

rPFS: progression-free survival rate

SBRT: stereotactic body radiation therapy

TILs: tumor infiltrating lymphocytes

TMB: tumor mutational burden

TRICOM: TRIad of CO-stimulatory Molecules

TRAEs: treatment-related adverse events

Moreover, we added some missing abbreviations in the text, including the following (green):

Prostate Specific Antigen (PSA) levels

Overall response rate (ORR)

Thank you again for your suggestions.

Reviewer 2 Report

As we know, tumor immunotherapies are emerging as a beneficial tool for cancer treatment by using our own immune system to prevent, control, and eliminate cancer. This manuscript summarize the current application status of immunotherapy in prostate cancer, including biomarkers of response, cancer vaccines, chief trials on immune checkpoint inhibitors and novel immune-based combinations. But there are also other published reviews about immunotherapy in prostate cancer recently, for example, “Past, Current, and Future of Immunotherapies for Prostate Cancer” which was published on Front. Oncol. in 2019; “Immunotherapy in Metastatic Castration-Resistant Prostate Cancer: Past and Future Strategies for Optimization” which was published on Current Urology Reports in 2019; “The evolving landscape of immunotherapy in advanced prostate cancer” which was published on Future Medicine in 2019; “Immunotherapy for castration-resistant prostate cancer: has its time arrived?” which was published on Expert Opinion on Biological Therapy in 2020. Although this review is the updated immunotherapies in PC, it is still lack of novelty.

I think our Cells journal can accept this manuscript if they do some major revisions to make this review stand out from many similar reviews.

  1. Although this review is about prostate cancer, this manuscript just focuses on prostate cancer, rarely describes the immunotherapy in other tumor patients. It would be better if this manuscript has a brief description of each immunotherapy in all tumor patients, not just prostate cancer. Readers may also want to know where prostate cancer is consistent and inconsistent with other tumors in immunotherapy. This will give the readers a more complete understanding of the role of immunotherapy in prostate cancer.
  2. Page 8, line 294-300, as the author mentioned, CA184-095 evaluated the safety and efficacy of ipilimumab, but the author just described the efficacy in this paragraph.
  3. This manuscript title is “Is there a role for immunotherapy in prostate cancer”. As we know, for the cancer immunotherapies, safety is as important as efficacy. Serious side effects can affect whether cancer drugs are approved for clinical use. There are a lot of patients will develop Immune-Related Adverse Events (irAEs) after immunotherapies, some patients have to stop immunotherapy if they have serious irAEs although they may benefit from the efficacy. For example, Tremelimumab, one anti-CTLA-4 antibody, has been undergoing human trials for the treatment of various cancers but has not attained approval for any. A lot of Tremelimumab clinical trials are terminated because of serious toxicity and unsatisfactory effectiveness. But this manuscript mainly focuses on the efficacy, rarely describes the safety.
  4. Page2, line 78, as this manuscript mentioned “the expression of PD-L1 does not appear to be a reliable biomarker of response to ICIs in PC”, one factor that cannot be ignored is that different labs maybe have different methods and scoring evaluation. The PD-L1 expressed on different cell types may have different functions.
  5. Page 11, table1, as the title mentioned “Ongoing clinical trials of immunotherapy in prostate cancer.” Why “NCT02814669” clinical trial which was completed is in this table? While other completed clinical trials of immunotherapy in PC are not included. If the author can make another table about the completed clinical trials of immunotherapy in PC would be better for the readers to know the clinical trial overview of immunotherapy in PC.
  6. Page 11, table 1, what is the standard for the clinical trials involved in table 1? Because there are a lot of ongoing clinical trials of immunotherapy in PC are not included. For example, NCT03532217, NCT02985957, NCT04109729, NCT02643303, NCT03385655, some of these unselected clinical trials have more PC participants with immunotherapies than listed clinical trials. Some of these clinical trials are also relevant and pertinent.
  7. Figures are easy to understand, but for figures 1 and 3, the symbol between T cells and tumors is usually thought as irradiation, not just killing. If the readers not read the figure legends carefully, they may confuse the irradiation symbol. The author may change it to other symbols or just use arrow and labeled with killing.

Author Response

Response to reviewer #2

Dear Reviewer, thank you for the time and effort you spent to provide your valuable feedback. We are aware that several papers (and in particular, narrative review articles) have been published in the last two years regarding immunotherapy in prostate cancer. We sincerely believe the manuscript has been improved and has gained in terms of originality, thank to your comments. In particular, we changes some details in Figure 1 and Figure 3 and we added twenty references regarding recent trials evaluating immunotherapy in distinct malignancies. Lastly, we included more safety details in trials exploring the role of immune checkpoint inhibitors in prostate cancer. We modified the manuscript according to your suggestions, as follows.

#1: Although this review is about prostate cancer, this manuscript just focuses on prostate cancer, rarely describes the immunotherapy in other tumor patients. It would be better if this manuscript has a brief description of each immunotherapy in all tumor patients, not just prostate cancer. Readers may also want to know where prostate cancer is consistent and inconsistent with other tumors in immunotherapy. This will give the readers a more complete understanding of the role of immunotherapy in prostate cancer.

#1: Dear Reviewer, thank you for pointing this out. We totally agree with your point of view since the development of immunotherapy has improved treatment outcomes for various types of cancer, with several agents which have been approved in the last five years. We added more details regarding recent trials resting immune checkpoint inhibitors in several malignancies, including metastatic malignant melanoma, renal cell carcinoma, non-small cell lung cancer and urothelial carcinoma. More specifically, we reported the details of Checkmate 214 trial, which has reported outstanding complete response rates in metastatic renal cell carcinoma, and some references to landmark trials regarding non-small cell lung cancer and malignant melanoma, where immunotherapy has undoubtedly modified the natural history of these malignancies.

In particular, we added the following part, introducing twenty additional references (purple). We modified as follows (purple):

The last decade has seen outstanding improvements in medical oncology, with the development and emergence of several novel agents and combinations [13-15]. Among these therapeutic approaches, a key role has been played by ICIs which have reported noteworthy results in a wide number of malignancies [16-18]. For example, medical treatment of metastatic melanoma, urothelial cancer, non-small cell lung cancer and renal cell carcinoma has been revolutionized in recent years, reporting unprecedented response rates and survival benefits [19-24]. Two meaningful examples are the impressive complete response rate of 10% achieved with nivolumab plus ipilimumab combination in metastatic renal cell carcinoma in Checkmate 214 trial and the survival benefits provided by ICIs in metastatic malignant melanoma where - until the approval of ipilimumab in 2011 – patients with distant metastases presented 5-year survival rates of approximately 5% [25, 26]. Moreover, on the basis of recent results of trials testing immunotherapy alone or in combination with other anticancer agents in different malignancies (e.g. gastric cancer, colorectal cancer, hepatocellular carcinoma, etc.) the number of indications for ICIs is supposed to further increase in the coming years [27-32]. However, if the advent of ICIs has certainly been a breakthrough in the therapeutic landscape of a number of hematological and solid tumors, the detection of specific molecular and histological biomarkers predictive of response to immunotherapy is the current challenge (…)

This topic is particularly important in malignancies where low response rates to ICIs have been observed so far, as in the case of PC (…)

  1. Mateo, J.; Lord, C.J.; Tutt, A.; Balmana, J.; Castroviejo-Bermejo, M.; Cruz, C.; Oaknin, A.; Kaye, S.B.; de Bono, J.S. A decade of clinical development of PARP inhibitors in perspective. Oncol. 2019, 30, 1437–1447.
  2. Motzer, R.J.; Escudier, B.; McDermott, D.F.; George, S.; Hammers, H.J.; Srinivas, S.; Tykodi, S.S.; Sosman, J.A.; Procopio, G.; Castellano, D.; et al. Nivolumab versus everolimus in advanced renal-cell carcinoma. Engl. J. Med. 2015, 373, 1803e13.
  3. Powles, T.; Smith, K.; Stenzl, A.; Bedke, J. Immune checkpoint inhibition in metastatic urothelial cancer. Eur. Urol. 2017, 72, 477–481.
  4. Pierantoni, F.; Maruzzo, M.; Gardi, M.; Bezzon, E.; Gardiman, M.P.; Porreca, A.; Basso, U.; Zagonel, V. Immunotherapy and urothelial carcinoma: An overview and future perspectives. Crit Rev Oncol Hematol. 2019, 143, 46-55.
  5. Ferris, R.L.; Blumenschein, G., Jr.; Fayette, J.; Guigay, J.; Colevas, A.D.; Licitra, L.; Harrington, K.; Kasper, S.; Vokes, E.E.; Even, C.; et al. Nivolumab for Recurrent Squamous-Cell Carcinoma of the Head and Neck. N. Engl. J. Med. 2016, 375, 1856–1867.
  6. Bellmunt, J.; de Wit, R.; Vaughn, D.J.; Fradet, Y.; Lee, J.L.; Fong, L.; Vogelzang, N.J.; Climent, M.A.; Petrylak, D.P., Choueiri, T.K.; et al. Pembrolizumab as second-line therapy for advanced urothelial carcinoma. Engl. J. Med. 2017, 16, 1015–1026.
  7. Robert, C.; Long, G.V.; Brady, B.; Dutriaux, C.; Maio, M.; Mortier, L.; Hassel, J.C.; Rutkowski, P.; McNeil, C.; Kalinka-Warzocha, E.; et al. Nivolumab in previously untreated melanoma without BRAF mutation. N. Engl. J. Med. 2015, 372, 320–330.
  8. Rosenberg, J.E.; Hoffman-Censits, J.; Powles, T.; van der Heijden, M.S.; Balar, A.V.; Necchi, A.; Dawson, N.; O’Donnell, P.H.; Balmanoukian, A.; Loriot, Y.; et al. Atezolizumab in patients with locally advanced and metastatic urothelial carcinoma who have progressed following treatment with platinum-based chemotherapy: a single-arm, multicentre, phase 2 trial. Lancet. 2016, 7, 1909–1920.
  9. Borghaei, H.; Paz-Ares, L.; Horn, L.; Spigel, D.R.; Steins, M.; Ready, N.E.; Chow, L.Q.; Vokes, E.E.; Felip, E.; Holgado, E.; et al. Nivolumab versus Docetaxel in Advanced Nonsquamous Non-Small-Cell Lung Cancer. Engl. J. Med. 2015, 373, 1627–1639.
  10. Sharma, P.; Callahan, M.K.; Bono, P.; Kim, J.; Spiliopoulou, P.; Calvo, E.; Pillai, R.N.; Ott, P.A.; De Braud, F.; Morse, M.; et al. Nivolumab in metastatic urothelial carcinoma after platinum therapy (CheckMate 275): A multicentre, single-arm, phase 2 trial. Lancet Oncol. 2017, 18, 312–322.
  11. Massari, F.; Mollica, V.; Rizzo, A.; Cosmai, L.; Rizzo, M.; Porta C. Safety evaluation of immune-based combinations in patients with advanced renal cell carcinoma: a systematic review and meta-analysis. Expert Opin. Drug Saf. 2020 Aug 16. doi: 10.1080/14740338.2020.1811226.
  12. Hanna, KS. Updates and novel treatments in urothelial carcinoma. Oncol. Pharm. Pract. 2019, 25, 648-656.
  13. Motzer, R.J.; Tannir, N.M., McDermott, D.F.; Frontera, O.A.; Melichar, B.; Choueiri, T.K.; Plimack, E.R.; Barthelemy, P.; Porta, C.; George, S.; et al. Nivolumab plus Ipilimumab versus Sunitinib in Advanced Renal-Cell Carcinoma. N. Engl. J. Med. 2018, 378, 1277-1290.
  14. Feld, ; Mitchell, T.C. Immunotherapy in melanoma. Immunotherapy. 2018, 10, 987-998.
  15. El-Khoueiry, A.B.; Sangro, B.; Yau, T.; Crocenzi, T.S.; Kudo, M.; Hsu, C.; Kim, T.Y.; Choo, S.P.; Trojan, J.; Welling, T.H.R.; et al. Nivolumab in patients with advanced hepatocellular carcinoma (CheckMate 040): an open-label, non-comparative, phase 1/2 dose escalation and expansion trial. Lancet 2017, 389, 2492–2502.
  16. Wrobel, P.; Ahmed, Current status of immunotherapy in metastatic colorectal cancer. Int. J. Colorectal Dis. 2019, 34, 13-25.
  17. Kang, B.W.; Chau, I. Current status and future potential of predictive biomarkers for immune checkpoint inhibitors in gastric cancer. ESMO Open. 2020, 5, e000791.
  18. Terrero, G.; Lockhart, A.C. Role of Immunotherapy in Advanced Gastroesophageal Cancer. Curr. Oncol. Rep. 2020, 22, 112.
  19. Pinter, M.; Scheiner, B.; Peck-Radosavljevic, M. Immunotherapy for advanced hepatocellular carcinoma: a focus on special subgroups. Gut. 2020, gutjnl-2020-321702.
  20. Carlisle, J.W.; Steuer, C.E.; Owonikoko, T.K.; Saba, N.F. An update on the immune landscape in lung and head and neck cancers. CA Cancer J. Clin.2020, 10.3322/caac.21630.

#2: Page 8, line 294-300, as the author mentioned, CA184-095 evaluated the safety and efficacy of ipilimumab, but the author just described the efficacy in this paragraph.

#2: Thank you for this comment. We added important data regarding the safety of ipilimumab in the randomized phase III CA184-095 trial comparing ipilimumab versus placebo.

In particular, we added the following part (orange):

(…) According to this phase III trial, the only grade 3-4 treatment-related adverse event (TRAE) reported in the 15% of enrolled subjects receiving ipilimumab. Importantly, immune-related adverse events were reported in 31% of subjects in the ipilimumab arm and the 2% of patients treated with ipilimumab (n=9) died due to TRAEs.

(…)

#3: This manuscript title is “Is there a role for immunotherapy in prostate cancer”. As we know, for the cancer immunotherapies, safety is as important as efficacy. Serious side effects can affect whether cancer drugs are approved for clinical use. There are a lot of patients will develop Immune-Related Adverse Events (irAEs) after immunotherapies, some patients have to stop immunotherapy if they have serious irAEs although they may benefit from the efficacy. For example, Tremelimumab, one anti-CTLA-4 antibody, has been undergoing human trials for the treatment of various cancers but has not attained approval for any. A lot of Tremelimumab clinical trials are terminated because of serious toxicity and unsatisfactory effectiveness. But this manuscript mainly focuses on the efficacy, rarely describes the safety.

#3: Dear Reviewer, thank you for your suggestions. Since immune-related adverse events, such as skin reactions, thyroid dysfunction, colitis, pneumonitis, hepatitis, etc. represent extremely important events in patients treated with immune checkpoint inhibitors, which typically do not occur with conventional cytotoxic anticancer agents, we added more details regarding immune-related adverse events in several trials.

In fact, we added the following paragraphs (orange - see #2 – and blue)

In terms of safety, grade 3-4 events adverse events occurred in the 26% (n=101) of patients in the ipilimumab arm and the 3% (n=11) of the placebo group, with diarrhea representing the most common event (16% and 2% of patients in the ipilimumab arm and the placebo arm, respectively). The 1% of patients enrolled in the ipilimumab group (n=4) died because of toxic effects of the PD-L1 inhibitor.

(…)

(…) According to this phase III trial, diarrhea was the only grade 3-4 treatment-related adverse event (TRAE) reported in the 15% of enrolled subjects receiving ipilimumab. Importantly, immune-related adverse events were observed in 31% of subjects in the ipilimumab arm and the 2% of patients treated with ipilimumab (n=9) died due to TRAEs.

(…)

The 60.9% of patients presented TRAEs, the most frequent of which was nausea (13%); grade 3-4 TRAEs (peripheral neuropathy, lipase increase, fatigue and asthenia) were reported in the 17.3% of patients (n=4). Lastly, no treatment-related deaths or discontinuation occurred during this trial.

(…)

With regard to toxicity, the 60% of patients presented TRAEs, in the 15% of cases of grade 3, 4 or 5. Finally, the 5% of patients discontinued treatment due to pembrolizumab-related adverse events.

(…)

In terms of safety profile, TRAEs have been observed in the 77.8% and the 51.1% of patients receiving atezolizumab plus enzalutamide or enzalutamide monotherapy, respectively. Moreover, grade 3-4 TRAEs have been reported in 28.3% of patients treated with the immune-based combination and the 9.6% of enzalutamide arm. Lastly, grade 5 TRAEs occurred in 1.9 and 0.3% of enrolled subjects.

(…)

#4: Page2, line 78, as this manuscript mentioned “the expression of PD-L1 does not appear to be a reliable biomarker of response to ICIs in PC”, one factor that cannot be ignored is that different labs maybe have different methods and scoring evaluation. The PD-L1 expressed on different cell types may have different functions.

#4: Dear Reviewer, thank you for pointing this out. We added the following sentence, according to your suggestions (grey):

However, methods of PD-L1 evaluation may vary widely in distinct trials and across laboratories, with the presence of different assays and scoring systems to define the cut-off positivity for PD-L1, and thus, the first step before exploring the impact of PD-L1 would probably be to use a standardized, single method.

#5: Page 11, table1, as the title mentioned “Ongoing clinical trials of immunotherapy in prostate cancer.” Why “NCT02814669” clinical trial which was completed is in this table? While other completed clinical trials of immunotherapy in PC are not included. If the author can make another table about the completed clinical trials of immunotherapy in PC would be better for the readers to know the clinical trial overview of immunotherapy in PC.

#6: Page 11, table 1, what is the standard for the clinical trials involved in table 1? Because there are a lot of ongoing clinical trials of immunotherapy in PC are not included. For example, NCT03532217, NCT02985957, NCT04109729, NCT02643303, NCT03385655, some of these unselected clinical trials have more PC participants with immunotherapies than listed clinical trials. Some of these clinical trials are also relevant and pertinent.

#5 and #6: Dear Reviewer, thank you for your comments.

In effect, we double-checked all the included trials and we made some changes. In particular, we would like to delete the NCT02814669 clinical trial from the table titled (Ongoing clinical trials of immunotherapy in prostate cancer”. Moreover, we would kindly ask to maintain only the table regarding ongoing trials, in order to limit the number of tables with a long list of studies.

In addition, we added all the following trials (NCT03532217, NCT02985957, NCT04109729, NCT02643303, NCT03385655, green), in order to report a more comprehensive table with main features of all ongoing clinical trials assessing immunotherapy in prostate cancer.

#7: Figures are easy to understand, but for figures 1 and 3, the symbol between T cells and tumors is usually thought as irradiation, not just killing. If the readers not read the figure legends carefully, they may confuse the irradiation symbol. The author may change it to other symbols or just use arrow and labeled with killing.

#7: Thank you for this suggestion. We slightly modified the figures, adding some “killing” symbols which may better explain the action of T cells. Moreover, we added the words “and cell death.” (red) to the legend of Figure 3, to help readability and comprehension of the figure.

Thank you again for your precious observations.

Best regards

Round 2

Reviewer 2 Report

Dear authors,

Thank you very much for your manuscript modifications according to my suggestions.

I think you still need to do some modifications.

  1. Immune checkpoint inhibitors (ICIs) maybe not accurate to describe the monoclonal antibody to PD-1/PD-L1/CTLA-4 in immunotherapy, especially for antibody to CTLA-4. A growing body of literature (PMID: 23897981/29576375/ 29472691/25918390/29581255, et al) shows that depletion of intratumoral Treg by ADCC/ADCP is the MOA (mechanism of action) for Ipilimumab‘s immunotherapy effect. Tregs are the main cells with higher CTLA-4 expression, they are easy to be depleted by ADCC/ADCP; Ipilimumab’s blocking ability is very weak; Mutant CTLA-4 antibodies which loss ADCC ability but keep blocking ability will destroy the anti-tumor effect; Blocking the Fc/FcR interaction will destroy the immunotherapy effect of CTLA-4 antibodies; Higher blocking ability of CTLA-4 antibody will not improve the anti-tumor effect, for example, Tremelimumab with higher blocking ability than Ipilimumab failed in a lot of clinical trials. So you may need to change “Immune checkpoint inhibitors” to “immune checkpoint monoclonal antibodies” or other accurate words. Although “Immune checkpoint inhibitors” is very hot and popular in literature, it is not mean it is correct or accurate.
  2. I think “Immune-Related Adverse Events (irAEs) may be better than “treatment-related adverse events (TRAEs)” in some places of your manuscript. Because your review is about immunotherapy in prostate cancer and talked a lot about immunotherapy. Immunotherapy induced adverse events are different from other treatments induced adverse events. You should distinguish them.
  3. Page 9, line 318-319, “The 1% of patients enrolled in the ipilimumab group (n=4) died because of toxic effects of the PD-L1 inhibitor. Are you sure ipilimumab treated toxicity is because of the PD-L1 inhibitor?

Author Response

Dear Reviewer, thank you for the time spent to revise our work. We have modified the paper in response to your last comments. We hope the revised manuscript will better suit Cells.

#1. Immune checkpoint inhibitors (ICIs) maybe not accurate to describe the monoclonal antibody to PD-1/PD-L1/CTLA-4 in immunotherapy, especially for antibody to CTLA-4. A growing body of literature (PMID: 23897981/29576375/ 29472691/25918390/29581255, et al) shows that depletion of intratumoral Treg by ADCC/ADCP is the MOA (mechanism of action) for Ipilimumab‘s immunotherapy effect. Tregs are the main cells with higher CTLA-4 expression, they are easy to be depleted by ADCC/ADCP; Ipilimumab’s blocking ability is very weak; Mutant CTLA-4 antibodies which loss ADCC ability but keep blocking ability will destroy the anti-tumor effect; Blocking the Fc/FcR interaction will destroy the immunotherapy effect of CTLA-4 antibodies; Higher blocking ability of CTLA-4 antibody will not improve the anti-tumor effect, for example, Tremelimumab with higher blocking ability than Ipilimumab failed in a lot of clinical trials. So you may need to change “Immune checkpoint inhibitors” to “immune checkpoint monoclonal antibodies” or other accurate words. Although “Immune checkpoint inhibitors” is very hot and popular in literature, it is not mean it is correct or accurate.

#1. Dear Reviewer, thank you for your suggestions.

In effect, we are aware that – although widely used in literature and modern “scientific communication” in medical oncology – the term immune checkpoint inhibitors would be a bit inaccurate to define the mechanism of action of CTLA-4 antibodies. In particular, we recognize the term would be inappropriate in Page 2, line 76, where we comment the noteworthy results provided by ipilimumab in malignant melanoma in the last years.

More specifically, as correctly stated in your comments, the action of CTLA-4 antibodies involves blockade of CTLA-4, causing an enhanced antitumor effector T cell activity. In this process, a key element is represented by the selective depletion of Treg cells and the role of Fc/FcR interaction.

In order to avoid misunderstandings, we changed ICIs to “immune checkpoint monoclonal antibodies”. However, since the term immune checkpoint inhibitors has been used during our search process and the studies selection, we kept the term in Page 2, line 63 – as you could notice.

Lastly, we modified throughout the manuscript, according to your suggestions (green):

Page 1, lines 28, 32

Page 2, lines 48, 50, 53, 54 59, 69, 76, 80, 82, 85, 91

Page 3, lines 100, 131, 143

Page 4, lines 151, 154, 169, 172, 193, 198

Page 5, lines 206, 208, 211

Page 7, line 267

Page 10, line 411

Page 11, line 427

------------------------------------------------------------------------------------------------------------------------------------------------

#2. I think “Immune-Related Adverse Events (irAEs) may be better than “treatment-related adverse events (TRAEs)” in some places of your manuscript. Because your review is about immunotherapy in prostate cancer and talked a lot about immunotherapy. Immunotherapy induced adverse events are different from other treatments induced adverse events. You should distinguish them.

#2. Thank you for pointing this out. In effect, we acknowledge that – by increasing the activation of the immune system – immunotherapy can cause a wide spectrum of side effects, better known as irAEs, which are specific of this therapeutic approach. We double-checked the manuscript in order to better distinguish among irAEs and TRAEs, changing the following parts, according to your comments (red):

Page 9, lines 337, 338, 339, 346, 347, 363

Page 10, lines 370, 380, 397,

However, in trials investigating immune-based combinations (including the KEYNOTE-199, the IMbassador250, etc.) we would prefer to use the term TRAEs since these studies assessed the combination of PD-L1 or PD-1 inhibitors with anticancer agents having different mechanisms of action (e.g. docetaxel, enzalutamide, etc.).

Finally, we added the abbreviation “irAEs” in the abbreviations list (Page 16, line 498).

------------------------------------------------------------------------------------------------------------------------------------------------

#3. Page 9, line 318-319, “The 1% of patients enrolled in the ipilimumab group (n=4) died because of toxic effects of the PD-L1 inhibitor. Are you sure ipilimumab treated toxicity is because of the PD-L1 inhibitor?

#3. Thank you for catching this inaccuracy, which we have now corrected in the text – as follows (green):

(…) Four toxic deaths were reported in the ipilimumab group (1% of patients). (…)

Thank you again for your precious observations.

Best regards